# Evolution of Thermal Microcracking in Refractory ZrO_2_-SiO_2_ after Application of External Loads at High Temperatures

**DOI:** 10.3390/ma12071017

**Published:** 2019-03-27

**Authors:** René Laquai, Fanny Gouraud, Bernd Randolf Müller, Marc Huger, Thierry Chotard, Guy Antou, Giovanni Bruno

**Affiliations:** 1Bundesanstalt für Materialforschung und-prüfung (BAM), Unter den Eichen 87, D-12200 Berlin, Germany; rene.laquai@bam.de (R.L.); giovanni.bruno@bam.de (G.B.); 2Centre Européen de la Céramique, University of Limoges, 12 Rue Atlantis, 87068 Limoges, France; fanny.gouraud@gmail.com (F.G.); marc.huger@unilim.fr (M.H.); thierry.chotard@unilim.fr (T.C.); guy.antou@unilim.fr (G.A.); 3Institute of Physics and Astronomy, University of Potsdam, Karl-Liebknecht-Str.24-25, 141176 Potsdam, Germany

**Keywords:** electro-fused zirconia, microcracking, synchrotron x-ray refraction radiography (SXRR), thermal expansion

## Abstract

Zirconia-based cast refractories are widely used for glass furnace applications. Since they have to withstand harsh chemical as well as thermo-mechanical environments, internal stresses and microcracking are often present in such materials under operating conditions (sometimes in excess of 1700 °C). We studied the evolution of thermal (CTE) and mechanical (Young’s modulus) properties as a function of temperature in a fused-cast refractory containing 94 wt.% of monoclinic ZrO_2_ and 6 wt.% of a silicate glassy phase. With the aid of X-ray refraction techniques (yielding the internal specific surface in materials), we also monitored the evolution of microcracking as a function of thermal cycles (crossing the martensitic phase transformation around 1000 °C) under externally applied stress. We found that external compressive stress leads to a strong decrease of the internal surface per unit volume, but a tensile load has a similar (though not so strong) effect. In agreement with existing literature on β-eucryptite microcracked ceramics, we could explain these phenomena by microcrack closure in the load direction in the compression case, and by microcrack propagation (rather than microcrack nucleation) under tensile conditions.

## 1. Introduction

The manufacturing of high quality glasses required for new applications (e.g., flat LCD or PDP screens) imply the development of new, high zirconia fused-cast refractories with excellent thermomechanical properties [1]. In order to build suitable refractory linings for glass furnaces, this ‘high zirconia’ (meaning high zirconia content) material is typically cast in heavy prismatic industrial blocks (about 1 m^3^) that are then adjusted to build the inner wall (lining). During the controlled cooling step after casting at about 2500 °C, dendrites of zirconia initially grow under the form of cubic domains (C), but then transform into tetragonal domains (T) at around 2300 °C. Upon further cooling, between 1000 °C and 900 °C, zirconia transforms from a tetragonal to a monoclinic (M) crystal structure. This transformation is associated with a large volume expansion of about 5% [2]. This generates large local (micro) stresses and typically microcracks [3]. While the polymorphism in zirconium ceramics is very well known [4,5], most of the literature is limited to the case of structural ceramics utilized in a rather low temperature range. In the present case, a high zirconia refractory material is targeted to very high temperature applications (typically 1500 °C, and even in excess of 1700 °C) within industrial furnaces for the production of ‘special’ glasses. In such a case, the stabilized (or partially stabilized) zirconia would be inadequate, since the refractory material is used above the zirconia phase transition (taking place between 1000 °C and 1160 °C in pure zirconia). Instead, a small quantity of glassy phase (SiO_2_) within the refractory microstructure is introduced to facilitate the industrial processing of large blocks (not otherwise possible). The glassy phase accommodates internal stresses induced by the large volume expansion associated to the zirconia phase transition during cooling (and also during preheating to the service temperature in the industrial furnace).

During the cooling stage, refractory blocks are also submitted to thermal gradients between their core and their skin. Thereby the martensitic transformation of zirconia usually occurs under a significant thermal gradient that generates additional macro-stresses, which can further significantly modify local microcracking mechanisms and induce macrocracks.

Thermally-induced microcracking (in the following simply thermal microcracking) generally occurs in ceramics with anisotropic coefficient of thermal expansion (CTE) when cooling from the sintering temperature [6]. Thermal microcracking distinguishes itself from mechanical microcracking, since it is: (a) Reversible and (b) mostly insensitive to the thermal cycle history [6,7]. Typical materials undergoing thermal microcracking are aluminum titanate [7,8], cordierite [8] and β-eucryptite [9,10]. The same phenomenon can occur in composites, where the (brittle) constituents have different CTE. This phenomenon is visible in the hysteresis of both the thermal dilatation (and of the CTE) and of the Young’s modulus as a function of temperature [11]. In the present study, high zirconia fused-cast refractory are typically constituted by pure ZrO_2_ embedded in a silica-rich glassy phase (though present in very small amounts). In this case, the network of microcracks stems from the 5% volume expansion associated to the T → M transformation in the 1000 °C to 900 °C range, but also from the anisotropic thermal expansion coefficient of the monoclinic (M) phase below 1000 °C. This network can be significantly modified by the additional macro-stress field induced by the thermal gradient during the whole cooling process. Thus, on one hand, industrial cooling condition of the blocks could greatly influence local microcracking, and on the other hand this network of microcracks could also significantly modify the thermo-mechanical properties (CTE, thermal conductivity, elastic properties). Depending on the microstructural features (dendrites and domains size), microcracks could play a key role in the appearance of macrocracks, which would render the block unusable. This is why a better understanding of the evolution of microcracking under operating conditions (especially thermal cycles, as well as mechanical loads) is of particular interest for both fundamental and industrial aspects. Due to the inhomogeneity of thermal stresses (large differences between core and periphery), we needed to reproduce the effect of the local stress field undergone by the materials during annealing and, more particularly, during the phase transition. Therefore, we reported that specimens were submitted to different levels of uniaxial load (tensile or compressive) during the phase transition (T → M) of zirconia. In this way, the occurrence of a TRansformation Induced Plasticity (TRIP) phenomenon during the zirconia phase transition could be characterized (such phenomenon is extremely poorly documented in the literature [12,13]). Since one current hypothesis is that the applied stress generates additional damage through microcracking, and both density and orientation of such microcracks would influence the ‘plastic’ deformation of the transition, the main target of the present work was to monitor the effect of thermal history (thermal cycling with/without external uniaxial loading) on the network of microcracks and on some thermomechanical properties (since a uniaxial load potentially leads to some anisotropy of such properties).

With this aim, we used novel techniques such as X-ray refraction radiography. The application of X-ray refraction [14] as a microstructure characterization technique at a macroscopic scale has proven to provide answers to questions that cannot be tackled even by the highest resolution techniques such as computed tomography. This is because X-ray refraction techniques possess detectability of features (cracks, pores, etc.) with size (~1 nm) well below their spatial resolution (~1 μm in the best case) as well as that of computed tomography. This detection power has been exploited in previous works using X-ray refraction [15,16], whereas model has been elaborated to rationalize the evolution of a network of microcracks in terms of propagation of large microcracks that could lead to the closure of smaller ones.

We will see below that microcracks evolved as a function of mechanical load, and are retained at the end of the thermal cycles. Such microcracks also engender a change of the equivalent elastic constants, therefore impacting the materials properties.

It belongs to future work to embed this damage behavior in new elasto-plastic constitutive laws.

## 2. Materials and Methods

### 2.1. Specimen Manufacture

The studied material was a fused-cast refractory containing 94 wt.% of monoclinic ZrO_2_ and 6 wt.% of a silicate glassy phase. Materials were typically obtained under the form of large blocks (1 m^3^) by a melting process in arc furnaces followed by controlled cooling in ceramic or graphite molds. All investigated specimens were machined by diamond tools from these cast blocks.

For investigations under a uniaxial load at a high temperature, specimens were prepared from cylindrical rods of 20 mm in diameter with metallic parts glued at each end. The final dog bone geometry was obtained by machining simultaneously the central zone (25 mm gauge length) of the specimen down to a diameter of 16 mm and the metallic parts. This allowed obtaining an optimized alignment with the loading axis.

In the second step, for X-ray refraction investigations, smaller prismatic specimens (30 mm × 16 mm × 0.5 mm) were machined in the central zone of the previous ones with a good identification of the load axis. For elastic properties measured by ultrasound in different directions at room temperature, cubic specimens (10 mm × 10 mm × 10 mm) cut with the same procedure, were used.

### 2.2. Microstructural Characterization

In order to investigate the complex microstructure of these high zirconia fused-cast refractories, coupons were cut and polished for SEM imaging. A Carl Zeiss AG-SUPRA 40 (Carl Zeiss AG, Oberkochen, Germany) with the following experimental conditions was used: Accelerating voltage = 15 kV and specimen-detector distance = 7 mm. A total of 4 material conditions were investigated: As-cast, after purely thermal cycling and after thermal cycling under tensile and compressive loads.

### 2.3. Dilatometry

A horizontal dilatometer (Netzsch DIL 402 C, NETZSCH-GERÄTEBAU GMBH, Selb, Germany) was used for thermal expansion analysis (no applied load). A large specimen (25 mm length) was prepared in order to reduce measurement error. As usual for such dilatometric experiments, a first test had to be performed on a calibration specimen (high purity sintered alumina), so as to determine and then subtract the specimen holder’s dilatation for further experiments on investigated materials.

### 2.4. Constrained Expansion Measurements

To reproduce the stress field undergone by the materials during the casting process (and during the phase transition), some specimens were submitted to different levels of uniaxial stress when undergoing the T → M transformation. These tests were performed with an INSTRON 8862 electro-mechanical universal testing machine (INSTRON, Norwood, MA, USA), which can operate up to 1600 °C. Strain was measured from the variation of a 25 mm gauge length obtained by 2 capacitive extensometers placed on the opposite faces of the specimen. The specimens were first heated up to 1500 °C with a rate of 10 °C/min, and then dwelled for 1 hour. This allowed them to return to a stress-free state. Then specimens were cooled with different temperature ramps, simulating the industrial production process of these materials. During cooling, the load was applied after a short dwell at 1150 °C. This ensured starting the application of the load at a low enough temperature to limit the risk of rupture, yet well above the T → M transformation. Once the load was applied, it was kept constant down to room temperature. A total of 2 tests were carried out with different stress levels (tension, +1 MPa and compression, −5 MPa). The values of the load were chosen, taking into account the strength values observed in tension and in compression at the temperature of the zirconia phase transition (about 1000 °C). We aimed at maximizing the effect of the TRIP phenomenon on the associated macroscopic deformation. In fact, at above 1.5 MPa tensile, rupture was systematically observed during the zirconia phase transition. The value of −5 MPa compressive was selected so that a more clearly visible effect on the deformation associated with the phase transition could be observed.

A reference cycle without any load (0 MPa) was run. As a side remark, we observed that the free dilation measured with the dilatometer and with the test rig, while having a similar trend, did not quantitatively match. This is due to the different calibrations of the 2 machines.

### 2.5. Ultrasonic Transmission

Ultrasonic measurements, in different directions, were carried out on specimens in the 4 conditions mentioned above. The velocities of longitudinal waves in infinite mode were measured according to the pulse transmission echo method [17]. The transmission method was applied due to the presence of some defects after thermal cycling under load, which can disturb the signal leading to a false measurement of the velocity if the reflection method were used.

A pulse generator/receiver system and 2 piezoelectric transducers (10 MHz) were used. A transducer was applied on 1 face of the specimen and directly sent ultrasonic waves across it. A second transducer collected the waves on the other face. The received signal was recorded on a digital oscilloscope. After signal analysis, the transit time through the thickness of the specimen was measured and related to wave velocity along a particular direction. Measurements were made in 3 perpendicular directions. The knowledge of the density allowed the calculation of the corresponding elastic constants.

### 2.6. X-ray Refraction

#### 2.6.1. Generalities about X-ray Refraction Techniques

X-ray refraction techniques were introduced a couple of decades ago [18] and have been successfully used for both material characterization and non-destructive testing [19]. X-ray refraction techniques are used to obtain the amount of the relative internal specific surface (i.e., surface per unit volume, relative to a reference state) of a specimen, and are therefore beneficial in the investigation of defects such as cracks and pores within ceramic components.

X-ray refraction occurs whenever X-rays interact with interfaces between materials of different densities as in the case of cracks, pores and particles in a matrix. The difference in the refraction indices between the 2 interfacing materials, the so-called refraction decrement, determines the refraction angle at the interface. The refraction decrement is dependent on the wavelength of the radiation. Since the refraction decrement for X-ray radiation is of the order of 10^−5^, X-ray optical effects can only be observed at very small scattering angles, which lie between a few seconds and a few minutes of arc. Since the typical X-ray wavelengths are approximately 0.1 nm, X-ray refraction detects pores and cracks as soon as they exceed a size of some X-ray wavelengths (so that the wave ‘notices’ the density difference at the interface). That means the smallest detectable object size is down to the nanometer range. This is not to be confused with the spatial resolution or the size of the objects that can be imaged. The spatial resolution is limited by the pixel size (in this work of about 4 μm × 4 μm) of the detector system. It must be emphasized that because of the inevitable background noise, it is impossible to conclusively detect one single defect. A certain population of objects is necessary to yield an integrated signal above the background noise. Thus, X-ray refraction is used primarily in radiographic mode with thin specimens (platelets) and yields a 2.5D signal, i.e., integrated over the specimen’s thickness. This results in the detection and imaging of a population of defects rather than the imaging of single defects. The X-ray refraction signal has been quantitatively correlated to microstructural changes and micromechanical models [20]. Furthermore, X-ray refraction techniques are sensitive to defect orientation, thereby allowing different kinds of defects to be identified [21]. The refraction signal of an isotropic inhomogeneity, such as spherical voids, will also be isotropic, whereas for cracks or elongated pores the signal vanishes when the defect surface normal is oriented perpendicular to the scattering vector of the detection system.

#### 2.6.2. Synchrotron Radiation X-ray Refraction

SXRR measurements were carried out at the BAM synchrotron laboratory BAMline at Helmholtz-Zentrum, Berlin, Germany [16,22]. The 3 specimens were mounted in a slide frame as shown in Figure 1. The beam energy was set to 50 keV (with ΔE/E ~0.2%) to achieve a specimen X-ray transmission of about 30%. A Princeton Instrument camera (2048 × 2048 pixel) in combination with a lens system and a 50 μm thick CWO scintillator screen provided a pixel size of 3.5 μm × 3.5 µm, capturing a field of view of about 7 mm × 3 mm [23].

A Si(111) analyzer crystal was placed in the beam path between the specimen and the camera system, shown in Figure 1, to perform refraction radiographs. The analyzer crystal reflect the beam coming out of the specimen into the detector system if the incidence angle is set to the Bragg angle, θ_B_ = 2.2664° at 50 keV. By tilting the analyzer crystal around an axis perpendicular to the scattering plane (the scattering plane of the Si(111) crystal lies in the plane of Figure 1), the so-called rocking curve is recorded. This describes the reflected beam intensity as a function of the deviation from the Bragg angle, Δθ = θ − θ_B_. The rocking curve was recorded for each specimen by taking 41 radiographs between θ = 2.2651° and θ = 2.2685° with a step size of Δθ = 0.0001° and exposure time of 5 s. All specimens were measured in two orientations: (a) Load direction of the specimen perpendicular and (b) parallel to the scattering plane (see Figure 1). In addition, the following images were acquired: Dark field (beam off) and flat field (beam on, but without specimens). The dark field image was used to subtract the dark current and detector readout noise from the specimen and flat field acquisitions. The flat field images quantified the instrumental artefacts and noise that were used to correct the X-ray radiography images according to Equation (1). The corrected rocking curve images were analyzed using an in-house software code based on LabView^®^. Figure 2 shows the typical rocking curves extracted from one arbitrary detector pixel. Open circles indicate the measurement without and filled circles with the specimen in the beam, respectively. The peak maxima are normed to unit to show the increase in beam divergence due to the refraction effect at interfaces (e.g., cracks and/or pores) inside the specimen. The analysis software delivered the values of the rocking curve integral, the peak height, peak position and the Full Width at Half Maximum FWHM (see Table 1). By using the image calculating software “Fiji Image J” [24] the attenuation (*µ*·*d*) and the refraction value (*C_m_*·*d*) were evaluated for each pixel according to Equations (1) and (2), respectively. A detailed description of the data processing and evaluation can be found in [25,26].
(1)$$\mu \cdot d=\ln\left(\frac{I_{0}}{I}\right)$$
(2)$$C_{m}\cdot d=1-\frac{I_{R}\cdot I_{0}}{I_{R0}\cdot I}$$

The influence of the specimens’ thickness *d* is eliminated by dividing the local refraction value (*C_m_*·*d*) by the local attenuation property (*µ*·*d*). This yielded the relative specific refraction value (*C_m_*/*µ*), which is proportional to the relative specific internal surface of the specimen up to a calibration constant, depending on the instrument, the material and the experiment (geometry and energy).

## 3. Results

### 3.1. Microstructure

Microstructural features associated with the cooling process are very complex. Figure 3a illustrates the different steps of microstructure evolution during the different stages of the cooling process [27]. At a very high temperature (2500 °C), dendrites of zirconia initially grew with a cubic structure (C). These dendrites possess primary and secondary ramifications (tree structure in Figure 3c); the structure transformed into tetragonal domains (T) at around 2300 °C. Between 2300 °C and 1700 °C, the mix (Figure 3b) was not supposed to be fully solid, and nucleation-growth of zirconia dendrites probably continued in this temperature range. Below 1700 °C, the material could be considered as fully solid with zirconia dendrites embedded in a silica glassy phase (Figure 3d). Between 1000 °C and 900 °C, the martensitic transformation of zirconia from the tetragonal to the monoclinic structure (M) occurred (Figure 3a). The cubic-to-tetragonal transformation was associated to a 45° rotation of the a- and b-crystal axes around the c-axis. This rotation induced the possible formation of three distinct crystallographic variants from one single cubic crystal. During the tetragonal-to-monoclinic transformation, it was possible to form 24 different crystallographic variants. The β angle (between a and b) differed thus from 90° (being close to 99°). At room temperature each zirconia dendrite was therefore constituted of different monoclinic variants (Figure 3e,f). Considering the anisotropy in thermal expansion along the different crystallographic axes in the monoclinic structure, these different crystallographic variants induced thermal mismatches and then potential microcracking between variants (Figure 3f). The glassy phase within the microstructure was assumed, because of its low viscosity at this temperature, to accommodate internal stresses induced by the anisotropic expansion mismatch between ZrO_2_ grains during this transformation.

### 3.2. Free and Constrained Thermal Expansion

As a reference without any external applied stress, a classic dilatometric experiment was carried out up to 1500 °C. This determined the characteristic temperatures for phase transformations of zirconia, as well as the average amplitude of the different effects in the unconstrained case (Figure 4). The general shape of this curve described an open hysteresis cycle with expansion discontinuities due to dimensional changes in the zirconia structure associated with transformations M ↔ T. The M → T transformation took place around 1115 °C, during heating (beginning of curve descent), whereas the inverse transformation occurred around 1010 °C, during cooling (beginning of sudden expansion). The amplitude of linear expansion associated with the phase transformations of zirconia depend on the zirconia content. In the present case, an expansion of 1.7% during cooling corresponded to a volume variation of 5.1%. This experimental value on high zirconia fused-cast refractories can be compared with the intrinsic volume variation associated with the transformation T → M in the case of a pure zirconia that can be measured by X-ray diffraction at high temperatures (3.6%) [12,28]. The macroscopic linear expansion of the present material was much larger than the microscopic (lattice) one, measured by XRD. The presence of a small amount of vitreous (silica) phase (6 wt.%, i.e., 12 vol.%), which has a slightly lower thermal expansion than zirconia, cannot explain this difference. This difference can only be explained by the variation of free space (void, microcracks) within the microstructure.

Results of the constrained dilation tests are presented in Figure 5, where strains are presented using 1500 °C as the reference state. It is clearly observed that the applied stress directly affected the strain associated with the tetragonal to monoclinic transformation. A tensile stress increased the strain associated to the T → M transformation, whereas a compressive one reduced it.

The ability of an external stress to influence the deformation associated with a phase transition has been already observed in metals [29].This phenomenon is known as Transformation Induced Plasticity. Thus, an interpretation of the present results can be based on similar mechanisms. The TRIP is a permanent macroscopic deformation occurring in the materials subjected to a phase transformation under an external mechanical stress even if this stress is much lower than the yield limit of the different phases that are present in the material. From a microstructural point of view, two mechanisms are usually considered to explain TRIP phenomenon in metals:The Greenwood–Johnson mechanism [30] corresponds to the micro-plasticity at the grain boundaries, which is required for the accommodation of the density differences during the phase transformation;The Magee mechanism [31] corresponds to a selective orientation of some crystallographic variants depending on the direction of the applied stress.

In the case of brittle material such as zirconia, a third mechanism corresponding to damage is also involved: Microcracking. The strain associated to microcracking is related thus to the number, the width and the orientation of microcracks generated under stress. The evolution of the microcrack density can be quantified by an elastic property measurement [8,9]. To this aim, Young’s modulus of all specimens was measured at room temperature before and after each test, applying a compressive stress between 0 MPa and 0.5 MPa. The relative changes of Young’s modulus after different cycles are reported below. A decrease in Young’s modulus was systematically observed after each test. In addition, this decrease was strongly correlated with the applied stress, being larger in the case of thermal cycles run under tensile stress. A thermal cycle run under compression induced a smaller decrease in Young’s modulus in comparison with an unconstrained thermal cycle.

### 3.3. Microcracking

Figure 6 shows the local linear attenuation coefficient *μ* as 2D color-coded images (the color spread is the same for all images). The 0 MPa and +1 MPa specimens show similar values of the linear attenuation coefficient. The attenuation of the 5 MPa specimens is about 5% higher. The attenuation is not homogeneous (areas of higher attenuation intersect with a network of lower attenuation). The spatial distribution of the attenuation is similar for all specimens, but the peak attenuation is higher for the 5 MPa specimen.

Figure 7 and Figure 8 show the local relative specific refraction value *C_m_*/*µ* as 2D color-coded images, respective for the load direction of the specimen perpendicular- and parallel-oriented to the scattering plane of the analyzer crystal (the color spread is the same for all images) for each specimen.

The 0 MPa specimen shows the highest specific surface in both orientations. The value of *C_m_*·*d* was roughly the same for both orientations (no preferred orientation of the features causing refraction, namely grain boundaries and microcracks). The areas of high specific surface are localized. 

The −5 MPa specimen shows the lowest specific surface. The value of *C_m_*·*d* was higher for the orientation of the load axis perpendicular to the scattering plane (preferred orientation of features parallel to the load axis). Also, the local maxima are higher than for the other two specimens. 

The 1 MPa specimen shows intermediate specific surface. The value of *C_m_*·*d* was roughly the same for both orientations (no preferred orientation). The local maxima are similar to the 0 MPa specimen.

### 3.4. Quantification of Elastic Constants’ Anisotropy

Since the arrangement of microcracks depends on the applied load during cooling (closure occurs in the case of compression, propagation in the case of tension), the degree of anisotropy of damage was quantified here through the measurement of the anisotropy of elastic constants (the microcrack induced anisotropy of properties has been predicted by Kachanov [32]). For this purpose, some entries of the stiffness tensor (*C*_ij_) were determined on specimens cooled under different applied stress through ultrasonic transmission measurements in different directions. A uniaxial load is likely to induce a transverse isotropic symmetry, therefore, measurements were focused on the constants *C*_11_, *C*_22_ and *C*_33_ (axis 3 is parallel to the applied load). For an isotropic material, the constants *C*_11_, *C*_22_ and *C*_33_ should be equal, whereas in the case of transverse isotropy, *C*_33_ would be different to *C*_11_ = *C*_22_. This anisotropy of the elastic constants was quantified (Figure 9b) through the index *AI*.
(3)$$AI=\frac{C_{33}}{\left[\frac{C_{11}+C_{22}}{2}\right]-1}$$

Figure 9 shows that:An unconstrained cooling yields a value of *AI* close to 0 (similar values of elastic constants in each direction), therefore to a rather isotropic microcrack arrangement;The application of a tensile stress during cooling leads to negative values of *AI* (*C*_33_ is smaller than *C*_11_ and *C*_22_). This implies the generation of a network of microcracks that are preferentially oriented in the plane perpendicular to the direction of application of the load;The application of a compressive stress during cooling leads to positive values of *AI* (*C*_33_ is larger than *C*_11_ and *C*_22_). This implies the generation of a network of microcracks that are preferentially oriented in the direction of the applied load.

In conclusion, these results suggest that the modulation of the deformation during the T → M transformation of zirconia under load could be related to the preferential direction of microcracks. This would establish a relationship between damage distribution (orientation) and the applied load during the T → M transformation that leads to the TRIP effect. In some other work, the orientation anisotropy of the microcrack arrangement has already been deduced (from lattice strain neutron diffraction measurements) and exemplarily observed by electron backscatter diffraction (EBSD) in porous Al_2_TiO_5_ by Bruno et al. [11]. Nevertheless, it was not possible in this case to clearly determine the preferential orientation of microcracks in SEM pictures, since their field of view was limited.

## 4. Discussion

It was expected that an external uniaxial stress should close microcracks oriented perpendicular to the load axis in the compression case, and open them in the tensile case. Correspondingly, we also expected that in the compression case, a possible rise of the opened microcracks in the direction parallel to the load axis would occur. Table 2, summarizing the X-ray refraction results, shows some expected results: In the parallel orientation of the load axis to the scattering plane, the decrease of refraction value in the −5 MPa specimen (with respect to the behavior of the 0 MPa specimen) corresponded to the decrease of microcrack density (or specific surface); in the perpendicular orientation of the load axis to the scattering plane, the (slight) decrease of refraction value for the +1 MPa specimen (with respect to the behavior of the 0 MPa specimen) corresponded to microcrack closure in the direction perpendicular to the load (Poisson’s effect as microcrack lips come together and fall below the detection limit of the technique). However, some apparently surprising trends also appeared: In the parallel orientation of the load axis to the scattering plane, a slight decrease of the refraction value in the +1 MPa specimen occurred, which corresponded to a decrease of microcrack density (or specific surface); in the perpendicular orientation of the load axis to the scattering plane, the (slight) decrease of refraction value for the −5 MPa specimen occurred, which corresponded to microcrack closure in the direction perpendicular to the load. These two effects cannot be explained by Poisson’s contraction (tension case) or expansion (compression case). A plausible explanation has been predicated in [15]: Damage in microcracked ceramics actually proceeds by propagation of existing microcracks, rather than by formation of new microcracks. This would imply that small microcracks can suddenly find themselves in the shielding zone of larger ones, thereby falling below the detection limit of X-ray refraction (~1 nm). Indeed, it has been shown in [15] that even under tension, some regions of a microcracked material undergoes local strain release, and when unloaded the detectable specific surface decreases even if the actual microcrack density (defined as *ρ* = 1/*V*·∑_i_
*a*_i_^3^, where *V* = investigated volume and *a*_i_ = radius of the i-th microcrack) increases, because of its cubic dependence on the crack size *a*. Furthermore, the 1 MPa specimen did possess a larger refraction value in the parallel orientation of the load axis to the scattering plane, i.e., a larger specific surface of microcracks oriented perpendicular to the applied load, but the amount of external tension is not enough to propagate existing microcracks to the same amount that a compressive stress of −5 MPa can do.

We also have to take into account that the quantitative analysis of the X-ray refraction maps of Figure 7 and Figure 8 strongly depends on the segmentation procedure utilized to extract the refraction value. By applying different masks to the images, one can obtain slightly different results. In Figure 10 it is shown that different masks are obtained with different methods (see [33,34]). Those masks yield slightly different refraction values, as summarized in Table 3. Table 3 shows a similar trend to Table 2, with one important exception: Specimens 0 MPa and 1 MPa do not differ much. This analysis would rather support the hypothesis that indeed specimen 1 MPa did not undergo enough deformation to induce significant microcrack propagation (further damage to the initial condition), but this is subject to future work.

## 5. Conclusions

We have confirmed that synchrotron X-ray refraction is a useful technique to determine the evolution of damage, especially in brittle (microcracked) materials. While classically limited to light materials, we have expanded the use of X-ray refraction to a high-density material such as electro-fused refractory zirconia. We have shown that one can change the amount of microcracking in this material, a ZrO_2_-SiO_2_ composite, by means of an externally applied uniaxial stress during the cooling branch of a thermal cycle: A compressive load will close microcracks perpendicular to the applied load. This change therefore caused the anisotropy of the microcrack orientation. Upon application of a tensile load during cooling, microcrack propagation seemed to take place, whereby small cracks virtually closed (i.e., they fell below the detection limit of X-ray refraction techniques), however the X-ray refraction data can also be interpreted so that the investigated tensile load may not have induced enough damage to be detected. To clarify the issue, further investigations are needed.

## Figures and Tables

**Figure 1 materials-12-01017-f001:**
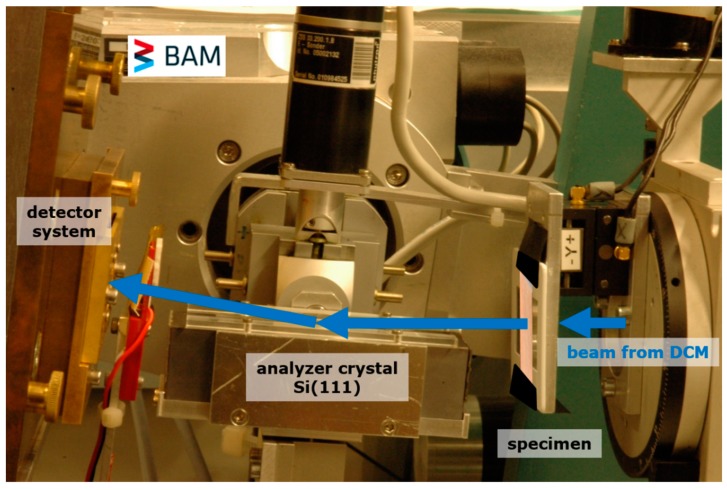
Experimental set up of the X-ray refraction station at BAMline. The specimens were mounted in a slide frame shown on the right. The scattering plane of the Si(111) crystal lies in the image plane.

**Figure 2 materials-12-01017-f002:**
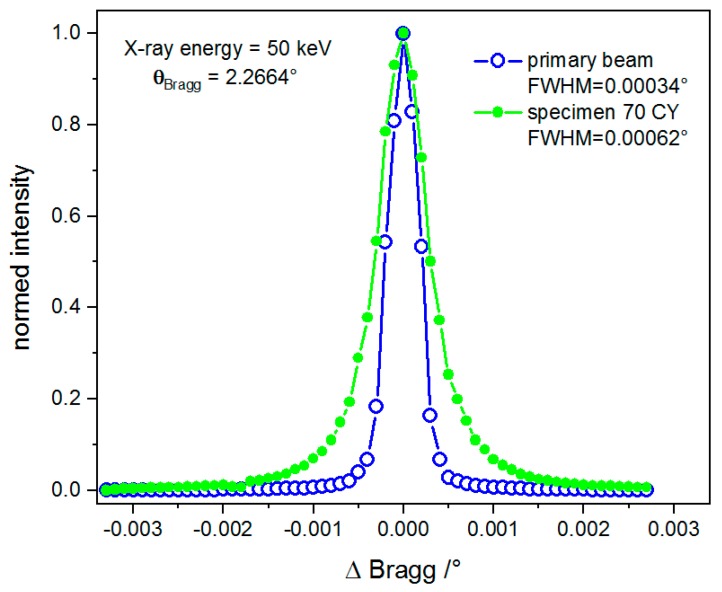
Rocking curves measured at one detector pixel and normalized to the peak maximum. Open circles: Without specimen (FWHM = 0.00034°); and filled circles: With specimen (FWHM = 0.00062°).

**Figure 3 materials-12-01017-f003:**
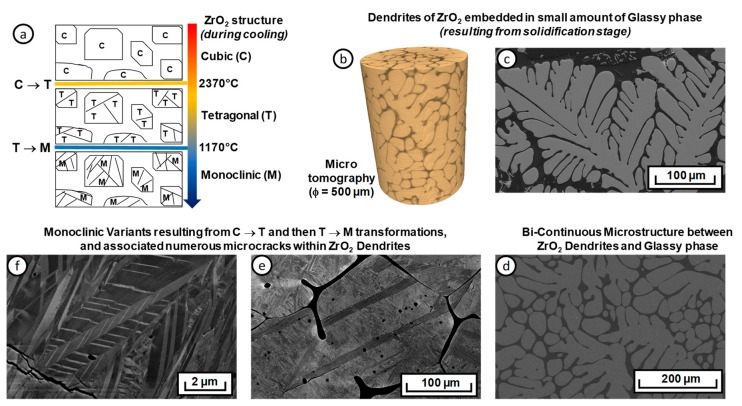
(**a**) Microstructure evolution of the ZrO_2_-SiO_2_ refractory; (**b**) 3D microtomography reconstruction of a cylindrical specimen, showing the dendritic structure, as in (**c**,**d**) (SEM pictures). (**e**,**f**) show the monoclinic variants stemming from the C → T and then T → M phase transformations.

**Figure 4 materials-12-01017-f004:**
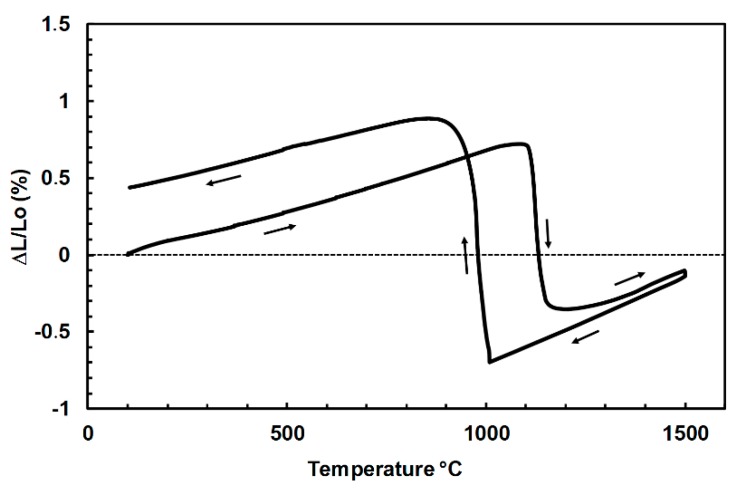
Unconstrained thermal expansion curve of the ZrO_2_-SiO_2_ refractory.

**Figure 5 materials-12-01017-f005:**
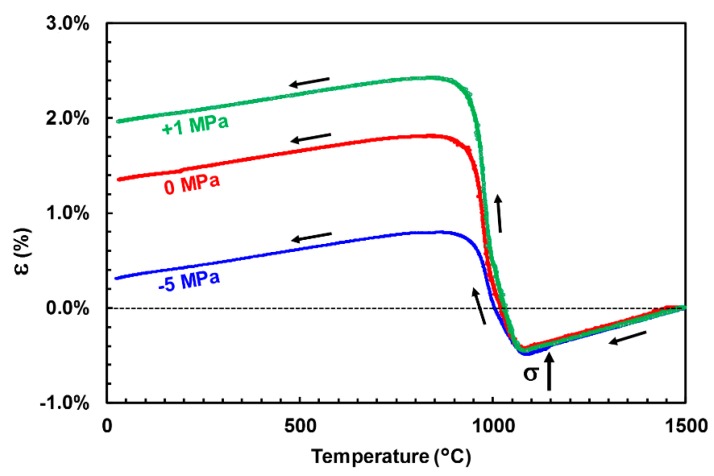
Constrained thermal expansion curve of the ZrO_2_-SiO_2_ refractory upon cooling, calculated assuming the strain-free state to hold at the maximum temperature (1500 °C). The stress was applied just before the start of the phase transformation (see vertical arrow).

**Figure 6 materials-12-01017-f006:**
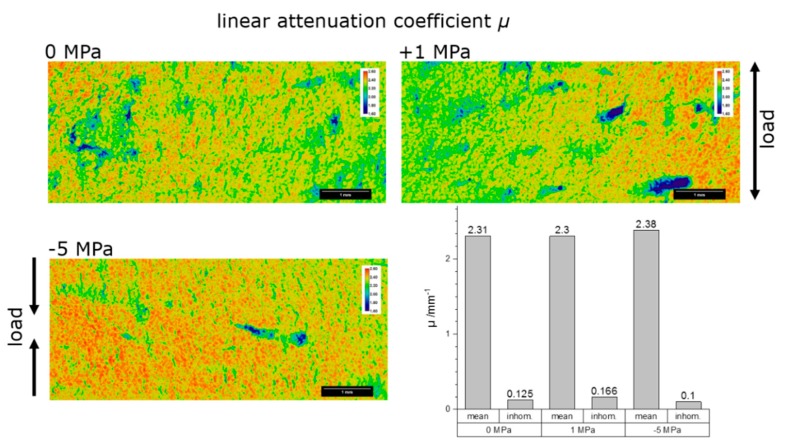
Visualization of the local values of the linear attenuation coefficient *µ* as 2D color-coded images (top left), unloaded (top right) 1 MPa tensile loaded and (bottom left) 5 MPa compression loaded. Bottom right: The integral values of the linear attenuation coefficient *µ* across the loaded and unloaded specimens are shown as bar graphs. The small bars represent the inhomogeneity of the *µ* values across the measured area of the specimen.

**Figure 7 materials-12-01017-f007:**
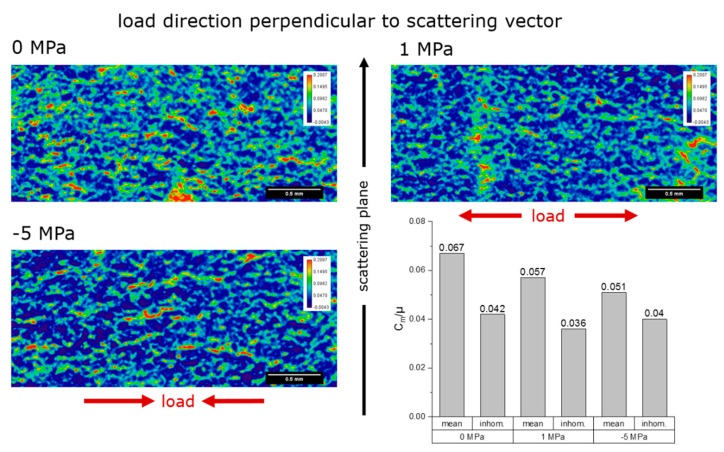
Visualization of the local values of the relative specific refraction value *C_m_*/*µ* of the specimen as 2D color-coded images. The load direction of the specimen was perpendicular to the scattering plane of the analyzer crystal. Top left: Unloaded, top right: 1 MPa tensile loaded and bottom left: 5 MPa compression loaded. Bottom right: The integral values of the relative specific surface for all specimens are shown as bar graphs. The small bars represent the inhomogeneity of the *C_m_*/*µ* values across the measured area of the specimen.

**Figure 8 materials-12-01017-f008:**
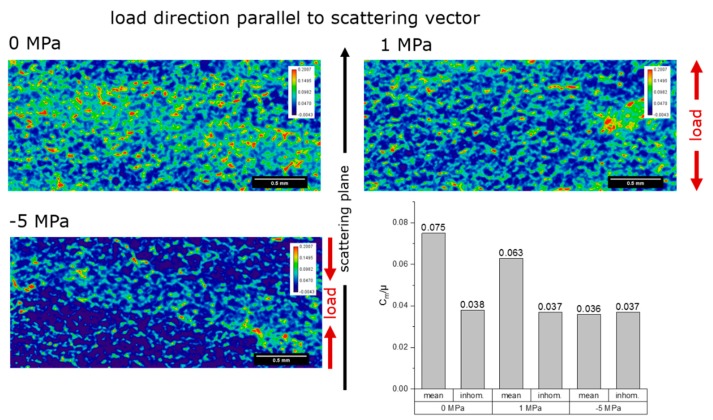
Visualization of the local values of the relative specific refraction value *C_m_*/*µ* of the specimen as 2D color-coded images for the orientation of the load direction of the specimen parallel to the scattering plane of the analyzer crystal. Top left: Unloaded, top right: 1 MPa tensile loaded and bottom left: 5 MPa compression loaded. The load direction was parallel to the scattering vector. Top left: Unloaded, top right: 1 MPa tensile loaded and bottom left: 5 MPa compression loaded. Bottom right: The integral values of the relative specific surface for all specimens are shown as bar graphs. The small bars represent the inhomogeneity of the *µ∙d* values across the measured area of the specimen.

**Figure 9 materials-12-01017-f009:**
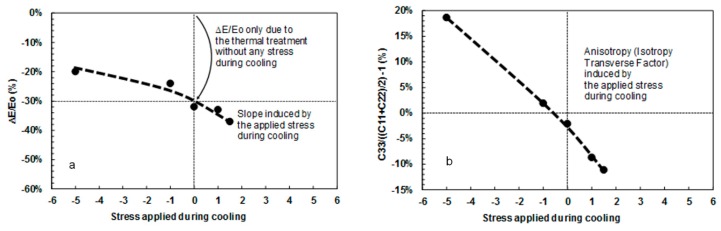
Room temperature elastic properties evolution due to thermal cycling (and cooling under applied stress, where axis 3 is the direction of the applied stress): (**a**) Decrease in Young’s modulus measured along the direction of the applied load and (**b**) evolution of the anisotropy index.

**Figure 10 materials-12-01017-f010:**
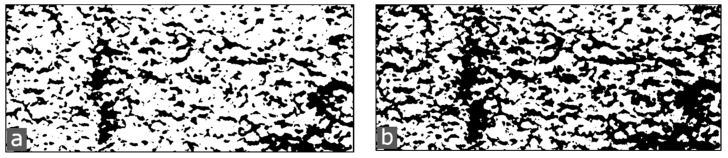
Segmentation of the refraction map for the 1 MPa specimen (Figure 7): (**a**) Mask with Otsu threshold; and (**b**) mask with Huang threshold.

**Table 1 materials-12-01017-t001:** The rocking curve parameter used to calculate the attenuation properties and the relative specific refraction value of the specimens.

Symbol	Quantity
*I_R_*	peak height (curve with filled circles) with specimen in the beam
*I_R_* _0_	peak height (curve with open circles) without specimen in the beam
*I*	peak integral (curve with filled circles) with specimen in the beam
*I* _0_	peak integral (curve with open circles) without specimen in the beam

**Table 2 materials-12-01017-t002:** Normalized global refraction value *C*_m_/*μ* as a function of orientation of the load axis of the investigated specimens to the scattering plane of the analyzer crystal. Relative error bars lie around 1–2%. Note that cracks perpendicular to the load axis are visible if the scattering plane is parallel to the load axis and vice-versa.

Specimen	Load Axis Perpendicular to Scattering Plane	Load Axis Parallel to Scattering Plane
+1 MPa	0.057	0.063
0 MPa	0.067	0.075
−5 MPa	0.051	0.036

**Table 3 materials-12-01017-t003:** Normalized refraction value *C_m_*/*μ* as a function of direction for the investigated specimens, calculated using two different segmentation masks (Otsu and Huang).

*C_m_*/*μ*	Otsu		Huang	
Specimen	Load Axis Perpendicular to Scattering Plane	Load Axis Parallel to Scattering Plane	Load Axis Perpendicular to Scattering Plane	Load Axis Parallel to Scattering Plane
+1 MPa	0.048	0.060	0.042	0.047
0 MPa	0.061	0.062	0.046	0.057
−5 MPa	0.038	0.030	0.037	0.032

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
