# Peer review of "Evolution of Thermal Microcracking in Refractory ZrO2-SiO2 after Application of External Loads at High Temperatures"

_materials, 2019, doi:10.3390/ma12071017_

Reviewer 1 Report

The research methods in the article are impressive. The authors convincingly showed their effectiveness in the study of ceramic materials under load.

The article needs minor corrections and additions.

The text of the article is displayed incorrectly, for example,

393    We also have to take into account that the quantitative analysis of the X-ray refraction maps of

394   Error! Reference source not found. and Error! Reference source not found. strongly depends on the

395   segmentation procedure utilized to extract the refraction value. By applying different masks to the

396   images, one can obtain slightly different results. In Error! Reference source not found. it is shown

397   that different masks are obtained with different methods (see [30,31]). Those masks yield slightly

398   different refraction values, as summarized in Error! Reference source not found.. Error! Reference

399   source not found. shows a similar trend to Error! Reference source not found., with one important

400    exception: specimens 0 MPa and 1 MPa do not differ much.

This complicates the perception of the material.

Similar things are found everywhere.

Despite the indicated difficulties of text perception, I would like to make the following general remarks.

The phenomenon of polymorphism in zirconium ceramics is well known and studied. It is well known that in order to stabilize the phase state, stabilizing additives, such as yttrium dioxide, are introduced into zirconium ceramics. In this regard, I would like the article to clearly identify the role of silicon dioxide. It is also necessary to explain why special additives were not introduced to obtain a stable phase state of the studied ceramics.

It would be well to clearly define which goal was pursued in this study. Is this a study of the properties of zirconium ceramics of a given composition under load, or the development of methods for monitoring the structure and properties of ceramics under load?

Thus, the reviewer recommends subjecting the article to revision.

Author Response

Sorry for the inconvenience concerning the figure cross references. Sometimes MS-Word does not work properly. We have deleted the automatic cross-references, and replaced them by hand. 

1- The phenomenon of polymorphism in zirconium ceramics is well known and studied. It is well known that in order to stabilize the phase state, stabilizing additives, such as yttrium dioxide, are introduced into zirconium ceramics. In this regard, I would like the article to clearly identify the role of silicon dioxide. It is also necessary to explain why special additives were not introduced to obtain a stable phase state of the studied ceramics.

We agree that the polymorphism in zirconium ceramics is very well known and has been very deeply studied, especially in the case of structural ceramics utilized in a rather low temperature range (taking great advantage of PSZ concepts). In the present case, this HZ refractory material is targeted to very high temperature applications (typically 1500°C) within industrial furnaces for the production of «special » glasses. In this case, the stabilized (or partially stabilized) zirconia is absolutely inadequate, since the refractory material is used above the zirconia phase transition (taking place between 1000°C and 1160°C in pure zirconia). In addition, for such refractory blocks in contact with the processed glass inside the furnace, the usage of conventional stabilizers is not recommended, because of potential dissolution of these stabilizers in the glass melt.

Typically, a small quantity of glassy phase (SiO2) within the refractory microstructure is introduced to facilitate the industrial processing of large blocks (not otherwise possible). In fact, the glassy phase accommodates internal stresses induced by the large volume expansion associated to the zirconia phase transition during cooling (and also during preheating to the service temperature in the industrial furnace).

 2- It would be well to clearly define which goal was pursued in this study. Is this a study of the properties of zirconium ceramics of a given composition under load, or the development of methods for monitoring the structure and properties of ceramics under load?

During the annealing step (cooling step), mechanical stresses are developed within the blocks. These stresses are inhomogeneous (large differences between core and periphery). They result from both thermal gradients and martensitic transformation of zirconia. To reproduce the local stress field undergone by the materials during the annealing and more particularly during the phase transition, specimens were submitted to different levels of uniaxial load (tensile or compressive) during the phase transition (TàM) of zirconia. Thanks to cooling tests under mechanical stress, the occurrence of a TRansformation Induced Plasticity (TRIP) phenomenon during the zirconia phase transition has been identified and characterized. Indeed, this TRIP phenomenon, in the case of zirconia transition, is extremely poorly documented in literature. One current hypothesis is that the applied stress generates an additional damage by microcracking where the number and/or the orientation of this cracks would influence the “plastic” deformation of the transition. Therefore, the present work aimed at casting light on this behavior. Indeed, we found that microcracks do evolve, and are retained at the end of the thermal cycles. It belongs to future work to embed this damage behavior in new elasto-plastic constitutive laws.

Reviewer 2 Report

This manuscript describes micro cracking evaluation in ZrO2- SiO2 under loads at high temperatures. Overall, it has been written by professional reaserchers and has a good organization.  My comments are as follows:

1- Please use use full name before abbreviations for the first time, for example CTE in abstract. 2- It is recommended to use some numbers in Abstract to give a better understanding for the readers.3- I think section 2 is too long and can be shorten. 4- In result section, there are some software problem “Error! Reference source not found”! 5- Why you have chosen 1 MPa and -5MPa tensions? 6- It is recommended to update reference by using some recent references instead of the older references ( before 2000).7-  Novelty of this manuscript should be explained in introduction.

Author Response

Reviewer 2

1.     Why you have chosen 1 MPa and -5MPa tensions?

The values of the load were chosen taking into account the strength values observed in tension and in compression at the temperature of the zirconia phase transition (about 1000°C). We aimed at maximizing the effect of the TRIP phenomenon on the associated macroscopic deformation. In fact, above 1.5 MPa tensile, rupture is systematically observed during the zirconia phase transition. The choice of -5 MPa compressive, a more clearly visible effect on the deformation associated with the phase transition could be observed.

2.     Please use full name before abbreviations for the first time, for example CTE in abstract.

Done

3.     It is recommended to use some numbers in Abstract to give a better understanding for the readers

Done, the abstract has been amended

4.     It is recommended to update reference by using some recent references instead of the older references ( before 2000).

Over 31 references in the original manuscript, only 7 are dated before 2000. Those are seminal works, to which one should always refer. A few other references have been added to make the points more clearly, but the most are older than 2000.

5.     Novelty of this manuscript should be explained in introduction.

This comment is linked to Reviewer’s 1 comment n.o 2. This point has also been integrated in the introduction. We report herewith the response to Reviewer 1.

During the annealing step (cooling step), mechanical stresses are developed within the blocks. These stresses are inhomogeneous (large differences between core and periphery). They result from both thermal gradients and martensitic transformation of zirconia. To reproduce the local stress field undergone by the materials during the annealing and more particularly during the phase transition, specimens were submitted to different levels of uniaxial load (tensile or compressive) during the phase transition (TàM) of zirconia. Thanks to cooling tests under mechanical stress, the occurrence of a TRansformation Induced Plasticity (TRIP) phenomenon during the zirconia phase transition has been identified and characterized. Indeed, this TRIP phenomenon, in the case of zirconia transition, is extremely poorly documented in literature. One current hypothesis is that the applied stress generates an additional damage by microcracking where the number and/or the orientation of this cracks would influence the “plastic” deformation of the transition. Therefore, the present work aimed at casting light on this behavior. Indeed, we found that microcracks do evolve, and are retained at the end of the thermal cycles. It belongs to future work to embed this damage behavior in new elasto-plastic constitutive laws.”

In result section, there are some software problem “Error! Reference source not found”!
Sorry for the inconvenience concerning the figure cross references. Sometimes MS-Word does not work properly. We have deleted the automatic cross-references, and replaced them by hand.